# Have Faith in Faithfulness: Going Beyond Circuit Overlap When Finding Model Mechanisms

**Michael Hanna[1],    Sandro Pezzelle[1],    Yonatan Belinkov[2]**
[1]ILLC, University of Amsterdam    [2]Technion—IIT, Israel
{m.w.hanna,s.pezzelle}@uva.nl   belinkov@technion.ac.il

## Abstract

Many recent language model (LM) interpretability studies have adopted the circuits framework, which aims to find the minimal computational subgraph, or *circuit*, that explains LM behavior on a given task. Most studies determine which edges belong in a LM's circuit for a task by performing causal interventions on each edge independently, but this scales poorly with model size. As a solution, recent work has proposed edge attribution patching (EAP), a scalable gradient-based approximation to interventions. In this paper, we introduce a new method—EAP with integrated gradients (EAP-IG)—that aims to efficiently find circuits while better maintaining one of their core properties: faithfulness. A circuit is faithful if all model edges outside the circuit can be ablated without changing the model's behavior on the task; faithfulness is what justifies studying circuits, rather than the full model. Our experiments demonstrate that circuits found using EAP-IG are more faithful than those found using EAP, even though both have high node overlap with reference circuits found using causal interventions. We conclude more generally that when comparing circuits, measuring overlap is no substitute for measuring faithfulness.

## 1 Introduction

The circuits framework (Olah et al., 2020; Elhage et al., 2021), a set of techniques from mechanistic interpretability aimed at localizing and explaining specific neural network behaviors, is increasingly applied to transformer language models (LMs). Most studies of circuits in LMs follow a simple structure: they use causal interventions to find components and connections that contribute to the behavior of interest, and then interpret the role of each component. Circuits have thus been used to explain how LMs predict indirect objects (Wang et al., 2023), complete year-spans (Hanna et al., 2023), track entities (Prakash et al., 2024), and more (Lieberum et al., 2023; Tigges et al., 2023; Merullo et al., 2024).

Unfortunately, circuit finding can be costly. Work on circuits often aims to identify not just important components (e.g., attention heads or MLPs) but also important edges, i.e. important connections between components. One can test an edge's importance by performing a causal intervention on it during a forward pass, and observing whether the relevant model behavior changes. However, LMs have many edges—GPT-2 small (Radford et al., 2019) has 32,491—so testing all edges is expensive, a problem that only grows with model size.

To combat this issue, Nanda (2023) introduced edge attribution patching (EAP), a gradient-based technique that can approximate the effects of causal intervening on each model edge in just two forward passes and one backward pass. Circuits found using EAP have high component overlap with those found manually (Syed et al., 2023), which might seem to like solid evidence that EAP works. Intuitively, if an EAP circuit overlaps highly with a known circuit, EAP has found the correct circuit. Other work, too, measures the success of circuit-finding via overlap (Conmy et al., 2023; Zhang & Nanda, 2024).

In this work, however, we focus on another quality that distinguishes circuits: *faithfulness*. We say a circuit is faithful to a LM's task behavior if ablating all of the LM's edges that are outside the circuit does not change its task behavior. Circuits' faithfulness is what justifies

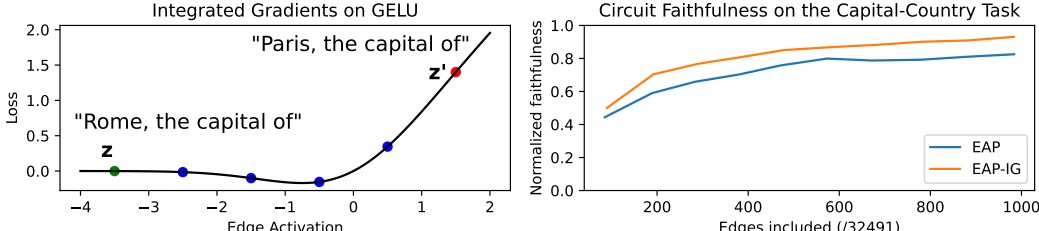

Figure 1: **Left**: Consider a circuit edge $e$ with activation $z$ on a chosen example, and $z'$ on a baseline. EAP assesses $e$'s importance via the loss' derivative at $z$, but this can be misleading: the flat derivative of the loss at $z$ indicates near-zero importance, but changes in $e$'s activation clearly cause changes in the loss. Inspired by integrated gradients (Sundararajan et al., 2017), we propose EAP-IG, which also considers the gradient at intermediate points between $z$ and $z'$. By computing better edge scores, EAP-IG finds more faithful circuits than EAP (**right**).

studying them: if our simplified model (the circuit) is unfaithful to the whole LM's behavior, studying it could yield misleading conclusions (Friedman et al., 2023). So, we ask: how faithful are EAP's circuits? And what is the relationship between overlap and faithfulness?

In this paper, we answer these questions, making three main contributions. First, we draw on earlier attribution literature (Sundararajan et al., 2017) to propose a new circuit-finding method—edge attribution patching with integrated gradients (EAP-IG)—that finds more faithful circuits (Figure 1). Second, we test the faithfulness of not only EAP and EAP-IG circuits, but also circuits found via activation patching scores, the ground truth that EAP and EAP-IG scores aim to approximate. We find that EAP-IG circuits are more faithful than EAP circuits on many tasks, though activation patching circuits still sometimes outperform both. Third, we empirically investigate the relationship between overlap and faithfulness, and show the former does not always imply the latter. We make the code for this paper's experiments available at `https://github.com/hannamw/eap-ig-faithfulness`; we also release an implementation of EAP-IG and other circuit finding methods at `https://github.com/hannamw/eap-ig`.

## 2   The Circuits Framework and Circuit Finding

The circuits framework (Olah et al., 2020; Elhage et al., 2021) seeks to reverse-engineer model behavior, by localizing it to subgraphs of the model's computation graph and explaining it.

**Circuits**   Recent work on circuits in transformer LMs (Wang et al., 2023; Hanna et al., 2023; Conmy et al., 2023, *inter alia*) generally considers a LM's **circuit** for a given task to be the minimal computation subgraph of the model whose behavior is faithful to the whole model's behavior on the task. Past work has used differing definitions of *computational graph*, *tasks*, and *faithful*, so we define these precisely here.

*Computational graph:* A transformer LM's computational graph is a digraph describing the computations it performs. It flows from the LM's inputs to the unembedding that projects its activations into vocabulary space. We define this digraph's nodes to be the LM's attention heads and MLPs, though other levels of granularity, e.g., neurons, are possible. Edges specify where a node's output goes; a node $v$'s input is the sum of the outputs of all nodes $u$ with an edge to $v$.[1] A circuit is a subgraph of this that connects the inputs to the logits.

*Tasks:* A circuits task comprises minimal pairs of clean and corrupted inputs, as well as a metric. Consider the task of subject–verb agreement: clean input might consist of strings like $s =$"The keys on the cabinet", which elicit a LM prediction like "are". Corrupted inputs fit the same task schema but elicit distinct output: e.g., $s' =$"The key on the cabinet" elicits

---

[1]In modern transformer LM architectures, a node's input is the sum of the outputs of all previous nodes; for details, see Elhage et al. (2021). Thus, any given node has edges to all downstream nodes.

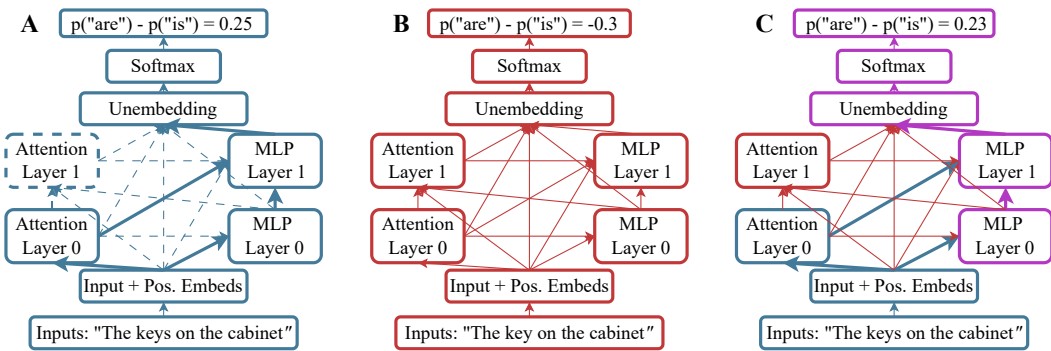

Figure 2: **A**: Computational graph of a model performing subject-verb agreement. Solid nodes and edges are in the task's circuit; dashed ones are not. **B**: The same model, run on the corresponding corrupted input. **C**: Testing circuit faithfulness, by replacing all non-circuit edges with corrupted activations from B. Red edges are corrupted; blue are not. Purple edges are the output of nodes receiving both corrupted and uncorrupted input. The circuit is faithful, as model behavior ($p$("are") - $p$("is")) stayed the same despite the intervention.

outputs like "is". This mirrors the approach of earlier behavioral (Linzen et al., 2016) and causal (Vig et al., 2020) interpretability studies. The metric, e.g. $M(x) = p(is|x) - p(are|x)$, measures the degree to which LM outputs reflect clean, as opposed to corrupted, input. The metric can be formulated differently, e.g. as a difference in logits or probability ratio (Wang et al., 2023; Vig et al., 2020), so long as it captures the relevant contrast in LM outputs.

*Faithfulness:* A circuit is faithful to model behavior on a task if we can corrupt all model edges outside the circuit while retaining the model's original task performance. That is, we run our model on clean inputs $s$ but replace the activations of edges not in the circuit with activations from the model when run on corrupted inputs $s'$; see Figure 2 for details.

More formally, we run the model on $s$, but perform the following intervention. Let $v$ be a node in the LM's computational graph, and $E_v$ be the set of all edges directed at it; let $i_e$ be a binary value indicating if an edge $e$ is in the circuit. Define $z_u$ as the output of a node $u$ during this run, and $z'_u$ as its output when the model is run on corrupted inputs.[2] Without interventions, the input to $v$ is $\sum_{e=(u,v)\in E_v} z_u$. To intervene, we set the input to each node $v$ to be: $\sum_{e=(u,v)\in E_v} i_e \cdot z_u + (1 - i_e) \cdot z'_u$. If all edges are in the circuit, this is identical to running the whole LM on clean inputs. If none are, it is equivalent to running it on corrupted inputs.

Typically, circuits are localized to a small portion of the LM's edges; the vast majority of the LM's activations are hence corrupted. However, if the circuit is correct, the model's behavior on the clean inputs should be roughly the same with or without these interventions. We verify this by computing our metric $M$ on the model with interventions. This should be close to $M$'s value when running the model without interventions on clean inputs.

We note that our particular test of faithfulness, which employs activation patching using corrupted examples, is not universal; others have used mean ablations over a corrupted dataset (Prakash et al., 2024), or even zero ablations (Olsson et al., 2022). Chan et al. (2022) advise against zero ablations, which may fall outside models' normal activation distribution, causing models to break even when unimportant nodes are ablated. Though they also advise mean ablations for the same reason, these seem in practice to be a reasonable alternative to patching, which we believe would yield similar results.

**Circuit Finding** Many studies identify circuits using activation patching (Vig et al., 2020; Geiger et al., 2021), which replaces a (clean) edge activation with a corrupted one during the model's forward pass. If the resulting change in the metric is greater than some threshold $\tau$, the edge is added to the circuit. Techniques like ACDC (Conmy et al., 2023) automate and

---

[2]Note that $z_u$ is *not* the activation of $u$ when our model is run on clean inputs without interventions; it is the activation of $u$ on the current forward pass, and $u$'s input may have been partially corrupted.

accelerate this process, but interventions fundamentally scale poorly with model size, with which both the cost of forward passes and number of edges grow. **Edge attribution patching (EAP)** (Nanda, 2023) ameliorates this. Given an edge $e = (u, v)$ with clean and corrupted activations $z_u$ and $z'_u$, EAP approximates the change in loss $L$ caused by corrupting $e$ as

$$(z'_u - z_u)^\top \nabla_v L(s), \tag{1}$$

where $\nabla_v L(s)$ is the gradient of $L$ with respect to the *input* of $v$. Note that EAP estimates the change in loss $L$ caused by intervening, rather than the change in metric $M$, but we can turn a metric into a loss by taking $L(x) = -M(x)$. The change in loss (or "score") for each edge can be computed with one forward pass on corrupted inputs, and one forward and backward pass on clean inputs. Having scored each edge, one then selects which edges fall in the circuit, e.g. by taking the top-$n$ edges by absolute score. In terms of node overlap, this method recovers manually found circuits with high precision and recall (Syed et al., 2023).[3]

## 3 EAP with Integrated Gradients

In this section, we design a new variant of EAP, taking advantage of the parallels between EAP and gradient-based *input attribution* methods. This body of work, which aims to determine which *input tokens* are most important to a model's behavior on a given example, has already analyzed and improved upon a method very similar to EAP: the baselined gradient $\times$ input method for input attribution (Shrikumar et al., 2017). The baselined gradient $\times$ input method scores an input token $t$'s importance as $(z_t - z'_t)^\top \nabla_t M(s)$, where $z_t$ is the token embedding of the $t$-th token of the input sentence $s$, and $z'_t$ is the same for some baseline $s'$. EAP's formulation, $(z'_u - z_u)^\top \nabla_v L(s)$, is essentially identical.

This similarity allows us to improve EAP by combining it with the integrated gradients (IG, Sundararajan et al., 2017) technique. IG aims to solve a problem that affects both the gradient $\times$ input method and EAP: zero gradients. Sundararajan et al. note that if of one of the model's internal activations has a zero gradient at $s$, that activation will not contribute to the attribution, even if the activation has a non-zero gradient at $s'$, and the difference in activations is significant; see Figure 1 for an example. IG resolves this issue by cumulating the gradients along the straight-line path from $s'$ to $s$. If a feature affects the predictions made on $s'$ as opposed to $s$, and our model is differentiable almost everywhere, there will be some point on this path where the feature has a non-zero gradient. The IG score for an input token $t$ is thus defined as the path integral of the gradients along the straight-line path from the baseline $s'$ to the input $s$; Sundararajan et al. approximate this via a sum as follows:

$$(z_t - z'_t) \int_{\alpha=0}^{1} \frac{\partial M(z' + \alpha(z - z'))}{\partial z_t} \approx (z_t - z'_t) \frac{1}{m} \sum_{k=1}^{m} \frac{\partial M(z' + \frac{k}{m}(z - z'))}{\partial z_t}, \tag{2}$$

where $m$ is the number of steps used to approximate the integral, $z$ is a sequence of token embeddings for one input, and $z'$ is the token embeddings of the distinct, baseline input.[4] We can then combine EAP and IG into a new method, **edge attribution patching with integrated gradients (EAP-IG)**.[5] The EAP-IG score of an edge $(u, v)$ is

$$(z'_u - z_u) \frac{1}{m} \sum_{k=1}^{m} \frac{\partial L(z' + \frac{k}{m}(z - z'))}{\partial z_v}. \tag{3}$$

---

[3]While we focus mainly on EAP, other recent work (Ferrando & Voita, 2024; Kramár et al., 2024) has introduced new methods for efficient circuit finding, though not using the lens of faithfulness. Due to their contemporaneity (and lack of publicly available implementation) we are unable to compare with them, but such comparisons would be valuable.

[4]Previously, we ran $M$ on string inputs ($s$ or $s'$); here, we specify the inputs as token embeddings, as IG runs our model on a blend of two inputs, which is possible only in embedding (not string) space.

[5]Contemporaneous work (Marks et al., 2024) has proposed a method similar to EAP-IG, which interpolates between node outputs, rather than input embeddings. This method is slower, as its cost scales with model depth, and achieves performance similar to that of our EAP-IG. See Appendix D for an in-depth comparison of the two methods and additional baselines.

EAP-IG allows for the use of losses like Kullback-Leibler (KL) divergence, which Conmy et al. (2023) recommend for use with activation patching because it is applicable to all tasks. For any task, an edge's importance can be measured by patching it, and observing the divergence between the patched LM's output distribution and the normal LM's output distribution. However, EAP estimates the effect of patching using gradient of the metric when the unpatched model is run on the clean input; in this case, the KL-divergence (and its gradient) will be zero. EAP-IG uses blended inputs, for which the gradient is non-zero.

EAP-IG has a runtime much like that of EAP. In theory, it is $m$ times slower, as it requires gradients for each of the $m$ steps. In practice, EAP-IG works with low values of $m$; we select $m = 5$ and justify this in Appendix C. Now, we must evaluate EAP-IG against EAP.

## 4 Evaluating Edge Attribution Faithfulness

How can we tell which of our two circuit finding techniques is better? Unlike earlier work, we forgo component overlap measures[6] in favor of comparing the faithfulness of the circuits found by EAP and EAP-IG, across six different tasks, on GPT-2 small (Radford et al., 2019). We also compare these to circuits based on scores from activation patching; recall that it is precisely these scores that EAP approximates. Computing these values is slow, but the faithfulness of these circuits represents the performance of EAP and EAP-IG with zero approximation error. Note, however, that activation patching circuits are imperfect; scoring each edge independently and then finding a circuit may be less accurate than e.g. ACDC.

### 4.1 Tasks

We test the faithfulness of EAP and EAP-IG on six tasks. We focus on simple tasks that are feasible even for GPT-2 small, the model most often studied from a circuits perspective. We intentionally choose some tasks (IOI, Greater-Than, and Gender Bias) that have been studied before; we omit others that are too hard for GPT-2 small (Prakash et al., 2024; Stolfo et al., 2023; Merullo et al., 2024). For more dataset and metric details see Appendix A.

**Indirect Object Identification (IOI)**: The IOI task (Wang et al., 2023) consists of inputs like "When Mary and John went to the store, John gave a bottle of milk to"; models are expected to predict "Mary". Their predictions are measured via logit difference (logit diff): logit of "Mary" minus the logit of "John". Corrupted inputs have the second instance of "John" changed to a third name, e.g. "Sally" leaving "John" and "Mary" roughly equiprobable. We generate a dataset of 1000 sentences using Wang et al.'s dataset generator.

**Gender-Bias**: The Gender-Bias task, designed to study gender bias in LMs, gives models input like "The nurse said that"; biased models tend to complete this sentence with "she". We measure bias via logit diff: the logit of "she" minus the logit of "he", or vice-versa, if the profession is male-stereotyped. Corrupted inputs replace female-stereotyped professions with "man" or male-stereotyped ones with "woman", leading the model to output the opposite pronoun. This task originates from Vig et al. (2020), and was studied in a circuits context by Chintam et al. (2023). We generate 2704 inputs using Vig et al.'s code, and filter it such that male- and female-biased professions are equal in number; this leaves 986 inputs.

**Greater-Than**: In the Greater-Than task (Hanna et al., 2023), models receive input like "The war lasted from the year 1741 to the year 17", and must predict a valid two-digit end year, i.e. one that is greater than 41. Model performance is measured via probability difference (prob diff): $\sum_{y=42}^{99} p(y) - \sum_{y=00}^{41} p(y)$. In corrupted inputs, the start year's last two digits are changed to "01", leading models to output years prior to the start year. Using Hanna et al.'s dataset generator, we create 1000 sentences, with varying events ("war") and start years.

**Capital–Country**: In the Capital-Country task, models receive input like "Tirana, the capital of" and must output the corresponding country (*Albania*). Corrupted instances contain another capital (e.g. *Brasilia*) instead. We measure performance using logit diff: the logit of

---

[6]EAP-IG does still successfully capture component overlap, though; see Appendix B

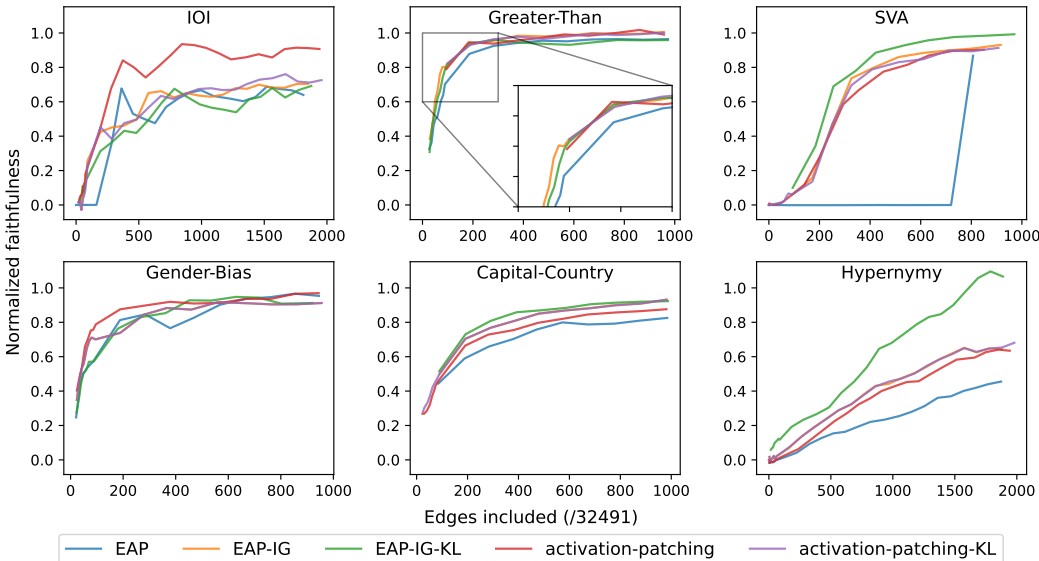

Figure 3: Faithfulness of circuits found via scores from EAP, EAP-IG, and activation patching; values near 1.0 are better. EAP-IG circuits' faithfulness equals or surpasses EAP circuits'.

the correct country (*Albania*) minus the logit of the corrupted country (*Brazil*). We create a Capital-Country dataset from the world's countries and capitals, with 190 examples.

**Subject-Verb Agreement (SVA)**: In SVA, models receive a sentence like "The keys on the cabinet", and must output a verb that agrees in number with the subject (*keys*), e.g. *are* or *have*. We measure models' ability to do so using prob diff, the probability assigned to verbs that agree with the subject, minus the probability of those that do not. In corrupted inputs, the subject's number is changed, e.g. from *keys* to *key*, causing the model to output verbs of opposite agreement. We sample 2000 such examples from Newman et al.'s (2021) SVA dataset, which contains varied sentence structures that might make SVA challenging.

**Hypernymy**: In the Hypernymy task, models must predict a word's hypernym, or super-ordinate category, given inputs like "diamonds, and other"; the correct answer is "gems" or "gemstones". Corrupted inputs contain an example of a distinct category, e.g. *cars*, which are *vehicles*. We measure performance with prob diff: the probability given to correct hypernyms, minus that of incorrect ones. We construct inputs using Van Overschelde et al.'s (2004) manually-collected set of hypernyms and instances thereof. This task is hard for small models, so we exclude inputs where GPT-2 small gets a prob diff $< 0.1$, keeping 198.

### 4.2 Methods

For each task, we score each model edge using EAP, EAP-IG, and activation patching. We convert each task metric $M$ (higher is better) to a loss $L$ (lower is better) by multiplying $M$ by -1; we also run EAP-IG and activation patching with KL divergence as the loss. Then, for $n = 30, 40, \ldots, 100, 200, \ldots, 1000$, we select a circuit of $n$ edges using a greedy search procedure; if no circuit is faithful at $n = 1000$, we consider up to $n = 2000$. We use greedy search as it produces more faithful circuits than sorting the edges by absolute score[7] and taking the top $n$ edges, as in Syed et al. (2023). For the full greedy algorithm, and details on finding circuits using scored edges, see Appendix E. In general, larger $n$ should yield circuits that are more faithful but less localized and interpretable. We aim to find circuits that are small (containing 1-2% of edges) and faithful (recovering $\geq 85\%$ of model performance).

---

[7]In both the greedy and top-$n$ cases, we select edges based on absolute scores. Selecting edges with the highest (non-absolute) scores would yield a better-performing circuit, but it would also discard components that are significant due to their *negative* effect on model performance.

Having found a circuit, we (recursively) prune all nodes and edges that have no children, or no parents. A node with no parent edges in the circuit would have all inputs corrupted, equivalent to not being in the circuit at all; childless nodes can likewise be discarded. Finally, we compute the circuit's performance, intervening on the model as described in Section 2, and applying the appropriate task metric. Because each task's metric has a different range, we create a normalized faithfulness score to enable cross-task comparison. Let $b$ and $b'$ be the whole model's performance on the clean and corrupted inputs of the given task respectively; we normalize the circuit's performance $m$ as $(m - b')/(b - b')$. We implement these and all following experiments in the TransformerLens library (Nanda & Bloom, 2022).

### 4.3 Results

Our results (Figure 3) show a wide variety of performance differences between EAP, EAP-IG, and activation patching. On IOI and Gender-Bias, both EAP and EAP-IG clearly under-perform activation patching circuits. On IOI, EAP and EAP-IG are much less faithful than activation patching (though notably only with logit diff, not KL divergence). The activation patching circuit quickly achieves faithfulness above 0.8, while EAP and EAP-IG circuits plateau at 0.6. In contrast, EAP and EAP-IG's underperformance on Gender-Bias is smaller and confined to $n \leq 500$; moreover, EAP and EAP-IG still achieve faithfulness over 0.8.

In Greater-Than and Capital-Country, there are clearer gaps between EAP-IG and EAP performance, and activation patching does not clearly dominate. In Greater-Than, the gap is small but significant: around 0.1 at the small $n$ that are most interesting for circuit finding. In Capital-Country, this difference is larger ($\geq 0.2$) at all values of $n$.

In SVA and Hypernymy, the gap between EAP and EAP-IG is large. In SVA, EAP displays a trademark flaw: it finds a completely unfaithful circuit until $n = 1000$. This is in part because EAP yields circuits with many parentless heads, which are then pruned.[8] At $n = 100$ and 200, EAP's circuits are pruned to nothing; even at $n = 1000$, EAP's circuit is pruned to 807 edges, (compared to 916 for EAP-IG, and 970 for EAP-IG-KL). All other methods avoid the traps into which EAP falls, by assigning a high score to the edge between the input embeddings and MLP 0, which is essential for high faithfulness.

Finally, in Hypernymy, the gap between EAP and the other methods is large and grows: by $n = 2000$, EAP-IG-KL has found a fully faithful circuit, while EAP's circuit is only halfway to faithfulness. However, neither circuit based on real patching scores achieves such high performance. As a result, there may be reason for caution— EAP-IG-KL might have found a circuit that achieves high performance, at the cost of missing high-importance heads that contribute negatively to Hypernymy. This, and the fact that neither EAP nor EAP-IG match activation patching for finding an IOI circuit, indicate that there is still room for improvement on top of EAP-IG. However, the faithfulness of circuits found with EAP-IG using the task metric still matches or surpasses that of EAP circuits.

### 4.4 Why Does EAP-IG Work?

Using the activation patching edge scores we obtained in our previous experiment, we investigate why EAP-IG often outperforms EAP. One might hypothesize that, in cases where EAP-IG outperforms EAP, it estimates activation patching scores better than EAP does. However, EAP-IG does not consistently have lower estimation error than EAP, and the difference between EAP and EAP-IG's total estimation errors is dwarfed by the scale of the errors themselves; see Appendix F for details EAP-IG scores are, however, better correlated with activation patching scores than EAP scores are. Figure 4 (left) shows that the Pearson correlation between activation patching scores and EAP-IG scores is consistently high (above 0.8), while the correlation of activation patching and EAP is always lower, though all correlations are significant ($p < 0.01$).

Despite this, EAP-IG does not better approximate the edge ranking induced by activation patching scores. Consider the ranking induced on edges by absolute activation patching,

---

[8]This is true when using either greedy search or thresholding to find circuits.

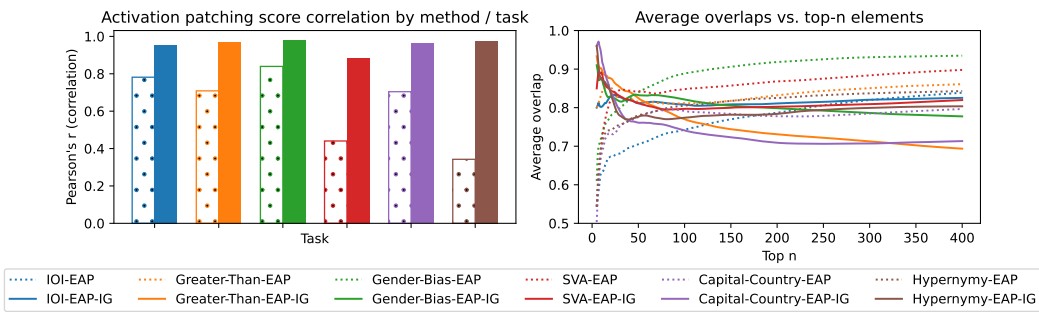

Figure 4: **Left**: Pearson correlation between the scores from EAP/EAP-IG and activation patching, by task. EAP-IG consistently has higher correlation. **Right**: Average overlap of top-$n$ edges of the activation patched circuit and the top-$n$ edges of EAP / EAP-IG circuits; higher is better. EAP-IG only generally has higher overlap only at small $n$ ($\leq 50$).

EAP, and EAP-IG scores. Activation patching's ranking has a significantly higher Kendall correlation (Appendix F) with EAP's ranking than EAP-IG's ranking. However, this measure is flawed: it considers ranking similarity across all edges, but most edges are near irrelevant for circuits. What matters is how EAP and EAP-IG rank the most important edges.

To quantify this, we compute the average overlap between the activation patched edge rankings and EAP / EAP-IG's edge rankings. The average overlap between two lists $X, Y$ at depth $n$ is the size of the intersection of their top-$n$ items, divided by $n$, i.e. $|X[: n] \cap Y[: n]|/n$. We compute this for $n = 1, 2, \ldots, 400$. Our results (Figure 4, right) show that while EAP-IG edge rankings do not generally have higher average overlap with those of manual patching, they overlap more at small $n$ ($< 50$). So, while EAP-IG's edge rankings are not overall more accurate, EAP-IG correctly scores performance-critical edges, yielding more faithful circuits.

## 5 Overlap and Faithfulness

Our previous experiments show that circuits found with EAP are not always faithful, despite their high node overlap with manually found circuits. This evidence (see Appendix G for more) pertains to within-task relationship of overlap and faithfulness, circuit but overlap can also be studied in cross-task scenarios (Merullo et al., 2024). So, we ask: to what extent does circuit overlap capture the similarity of mechanisms that models use across tasks?

To answer this, we analyze cross-task circuit overlap and faithfulness, i.e. the overlap and faithfulness obtained when comparing and running circuits across tasks. For each task, we take the smallest circuit we have found that achieves at least 85% faithfulness on the task; this entails using a circuit based on scores from real activation-patching or EAP-IG-KL. We then compute two overlap metrics: the intersection over union (IoU, also called Jaccard similarity) between the circuits' nodes, and the IoU between their edges. We also compute the statistical significance of each overlap; see Appendix H for details.

We then compute cross-task faithfulness: for every pair of circuits $C_1, C_2$, we run $C_1$ on $C_2$'s task (and vice-versa), recording the faithfulness achieved. We do under the assumption that if the model uses the same mechanisms to solve both tasks, the tasks' circuits should be faithful to one another. However, there is no guarantee that any pair of circuits will be faithful across our current set of tasks. This poses a problem: we cannot assess whether overlap predicts faithfulness, without any task pairs whose circuits exhibit cross-task faithfulness.

We resolve this issue by studying three additional tasks: Greater-Than (Price), Greater-Than (Sequence), and Country-Capital. The first two are simple variants on the Greater-Than task; they consist of inputs like "The ring's price ranges from $ 1741 to $17" and "1629, 1671, 1692, 1741, 17", respectively. Like Greater-Than, their corrupted inputs include a 01, and they use prob diff; for more details, see Appendix I. We include these tasks because Hanna et al. (2023) report that the Greater-Than circuit is both similar (though not identical) to these

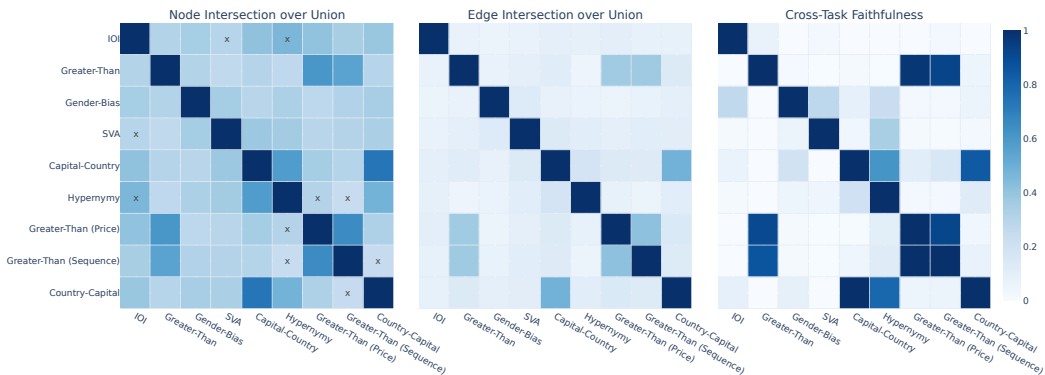

Figure 5: **Left** and **Center**: Node and edge intersection over union (Jaccard similarity) of circuits (85% faithful) for various tasks. All overlaps but those marked with an X are significant ($p < 0.01$, hypergeometric test). **Right**: Cross-task faithfulness. Each square is the faithfulness of the circuit from the y-axis task, tested on the x-axis task. Faithfulness is normalized, per test task, according to the faithfulness of the circuit for that task.

tasks' circuits, and generally faithful when run on them. The last task, Country-Capital, is the reverse of the Capital-Country task, consisting of examples like "Albania, whose capital", where the desired answer is "Tirana". We include this task because we hypothesize its circuit may overlap with that of Capital-Country.

The cross-tasks relationships suggested by each of our experiments are broadly similar, with small but key differences. Figure 5 (left) indicates that while all task pairs have at least a small level of node overlap (0.25 IoU), certain pairs have notably higher overlap. As expected, the Greater-Than task variants all have high overlap ($\geq 0.6$) with one other, as do the Capital-Country and Country-Capital tasks. However, the Hypernymy task also has high node IoU with the latter two tasks. This likely stems from the similar nature of the tasks; in both, the LM must retrieve a property of a given word. In Hypernymy, the LM retrieves the word's hypernym; in the latter pair, the LM retrieves the word's (city's) location or the word's (country's) capital.

In contrast with node overlap, the baseline edge overlap (Figure 5, center) between tasks is very small, often between 0.05 and 0.15 IoU. As in the node overlap scenario, the Greater-Than task variants, as well as Country-Capital and Capital-Country, have high edge overlap compared to other task pairs (0.4 IoU). However, this is not true of the Hypernymy and Capital-Country or Country-Capital tasks.

The pairwise faithfulness of tasks (Figure 5, right) is more complicated. As in the prior two scenarios, the Greater-Than task variants have high cross-task faithfulness, as do Capital-Country and Country-Capital[9]. However, while the Hypernymy circuit (*y*-axis) is highly faithful on the Capital-Country and Country-Capital tasks (*x*-axis), the reverse is not true. This is possible because faithfulness, unlike the other measures, which are based on IoU, is asymmetrical. As the hypernymy circuit is much larger than other circuits, it is likely better able to be faithful to a variety of tasks: it has moderate faithfulness on SVA and Gender-Bias as well. Most other task pairs have near-zero faithfulness.

This raises the question: to what extent does overlap inform faithfulness? The two are indeed strongly related: Pearson correlation of faithfulness with node overlap 0.86, and its correlation with edge overlap is 0.83; both are highly significant ($p < 0.001$). This is expected—component overlap must have some positive impact on faithfulness. A linear regression from node overlap to faithfulness achieves an $R^2$ of 0.74, while using edge overlap achieves 0.70, and combining the two yields $R^2 = 0.74$; these are all strong fits.

---

[9]This is an interesting result in light of the reversal curse: Berglund et al. (2024) find that LMs trained on "A is B" do not learn "B is A". Here, where the model has likely been trained on both facts, it uses similar mechanisms to retrieve both.

Despite this, we caution against concluding that overlap is a good way to predict faithfulness. The most interesting points are those where task overlap is moderate; it is unsurprising that circuits with near zero overlap are unfaithful across tasks, and that circuits with perfect overlap are. But when overlap is moderate, it does not predict faithfulness. The node overlap between Capital-Country and Hypernymy is 0.58, but the Hypernymy circuit achieves a faithfulness of 0.61 on Capital-Country; in the reverse case, faithfulness is 0.2. For one value of overlap, we have two very different values of faithfulness; no linear regression based on overlap can capture this. Even across tasks, there are challenges: the node overlap between Greater-Than (Price) and Greater-Than (Sequence) is 0.64, much like that of Hypernymy and Capital-Country, but cross-task faithfulness is high either way. We thus conclude that overlap is not a good predictor of cross-task faithfulness when overlaps are moderate.

## 6  Discussion

In this study, we have analyzed the question of faithfulness in circuit-finding, discovering that while EAP promised to automatically find circuits, its circuits' faithfulness was poor compared to EAP-IG's. We also found that while zero and full overlap predict low and high cross-task faithfulness, moderate overlap predicts little, leading us to recommend testing faithfulness separately from overlap. However, many open questions remain; here, we discuss these open questions and recommend best practices for circuit finding.

**Completeness** Faithfulness is not the only potential circuit desideratum: while measuring faithfulness ensures that our circuits stray little from whole-model performance, it does not guarantee that they are *complete*, i.e. not missing any important components, even those that inhibit model performance. Notably, while many circuits may achieve a given faithfulness, a circuit that is both faithful and complete should be relatively unique. We aimed to find complete circuits by selecting edges based on absolute score magnitude, thus including both positively and negatively important edges; this contrasts with other work, which prioritized finding positively important components (Prakash et al., 2024).

We would like to prove that EAP-IG's circuits are complete, but testing completeness is challenging. Wang et al. (2023) compare compare model and circuit behavior under random ablations, reasoning that a complete circuit should remain similar to the whole model even under arbitrary ablations. However, this is costly, and it is unclear how to choose sensible random ablations. If manually-found circuits are taken as ground-truth complete circuits, then circuit overlap could be considered metric of completeness, rather than faithfulness. Still, overlap as a metric is flawed: a completeness metric should ideally, like faithfulness, be measurable without a ground truth, as manually-found "ground-truth" circuits may be unreliable.

**Briding the gap between overlap and faithfulness** Overlap and cross-task faithfulness yield different results. This occurs in large part because the latter depends not on overlap generally, but on which specific nodes or edges overlap: for example, the edge from the input to MLP 0 is often crucial. This issue might be avoided with a metric, such as weighted graph edit distance, that takes into account each edge's importance (score). This would require published circuits to report edge scores, a practice that we would encourage.

**Best practices for circuits** As circuits become more popular and easier to find, practitioners' research questions increasingly involve comparing circuits across tasks and models. This is an important development, but as we attempt to answer such questions, we must be careful not to be misled about the actual mechanisms that underlie model behavior across tasks; these mechanisms are the core of circuits research. Measuring cross-task faithfulness and characterizing component behavior across tasks, will yield sounder results.

### Acknowledgments

The authors thank Neel Nanda for suggesting the Clean-Corrupted baseline and other comparisons. MH is partially supported by an OpenAI Superalignment Fellowship. This research has been supported by an AI Alignment grant from Open Philanthropy, the Israel

Science Foundation (grant No. 448/20), and an Azrieli Foundation Early Career Faculty Fellowship.

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

| Task | Metric | Clean Baseline | Corrupted Baseline |
|---|---|---|---|
| IOI | logit diff | 3.80 | 0.03 |
| Gender-Bias | logit diff | 0.88 | -3.22 |
| Greater-Than | prob diff | 0.81 | -0.46 |
| Capital-Country | logit diff | 6.12 | -6.72 |
| SVA | prob diff | 0.16 | -0.16 |
| Hypernymy | prob diff | 0.21 | -0.07 |
| Greater-Than (Price) | prob diff | 0.65 | -0.24 |
| Greater-Than (Sequence) | prob diff | 0.90 | -0.75 |
| Country-Capital | logit diff | 6.78 | -6.86 |

Table 1: Clean and Corrupted Baseline Performance of GPT-2 small across tasks

Kevin Ro Wang, Alexandre Variengien, Arthur Conmy, Buck Shlegeris, and Jacob Steinhardt. Interpretability in the wild: a circuit for indirect object identification in GPT-2 small. In *International Conference on Learning Representations*, 2023. URL `https://openreview.net/forum?id=NpsVSN6o4ul`.

Fred Zhang and Neel Nanda. Towards best practices of activation patching in language models: Metrics and methods. In *The Twelfth International Conference on Learning Representations*, 2024. URL `https://openreview.net/forum?id=Hf17y6u9BC`.

## A  Dataset and Metric Details

**Datasets**    Here, we provide additional details regarding our datasets. The Hypernymy task formulation was inspired by Hearst's (1992) work on automatically acquiring hypernyms from corpora. By using prompts that resemble contexts in which hypernyms naturally appear in data, we are able to elicit hypernym-retrieval behavior from even weaker language models. Similar strategies have been used by later work on hypernymy in masked language models (Ettinger, 2020; Ravichander et al., 2020; Hanna & Mareček, 2021). The Capital-Country task is also one that has been studied previously, though it has generally not the focus of any one work (see Appendix H of Merullo et al. (2024)).

Throughout this study, we report normalized faithfulness. However, in Table 1, we also report the unnormalized baseline performances of GPT-2 small on each task (either in logit diff or prob diff, according to the task). We also report the corrupted baseline (performance of the model given corrupted inputs).

**Metrics**    Our tasks are generally assessed with one of two metrics: logit difference (logit diff) and probability difference (prob diff). Logit diff (Wang et al., 2023) measures the difference in logits assigned to the correct and corrupted answer, e.g. $\text{logit}(\text{Mary}) - \text{logit}(\text{John})$ for IOI. However, logit diff extends poorly to multi-answer settings. Because logits are not normalized, we cannot simply take the difference in sums. One way to extend logit diff to the multi-answer domain is by taking the mean of the correct / incorrect logits, and taking the difference in means, but this is not entirely satisfying.

To avoid logit diff in the multi-answer setting, we can instead use prob diff (Hanna et al., 2023), which measures the difference in probability assigned to correct and incorrect answers. In Greater-Than, it sums the total probability assigned to all the correct and incorrect options, and takes the difference, e.g. $\sum_{y>41} p(y) - \sum_{y\leq 41} p(y)$. We do not use prob diff in the single-answer setting, because some prior work has reported that it can hide the role of components that are important to the model behavior in a negative way, reducing metric value (Zhang & Nanda, 2024).

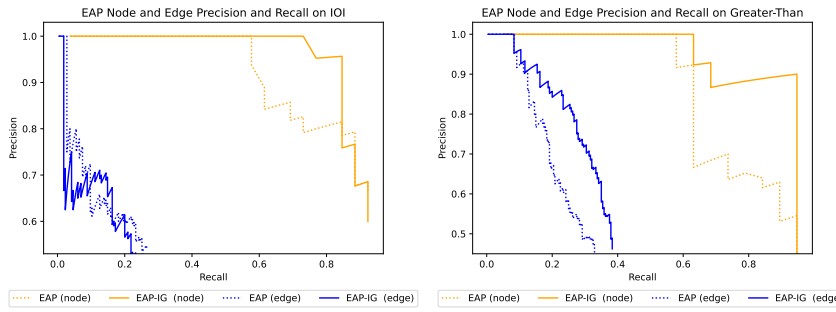

Figure 6: Precision-recall curves for IOI (left) and Greater-Than (right) node / edge overlap

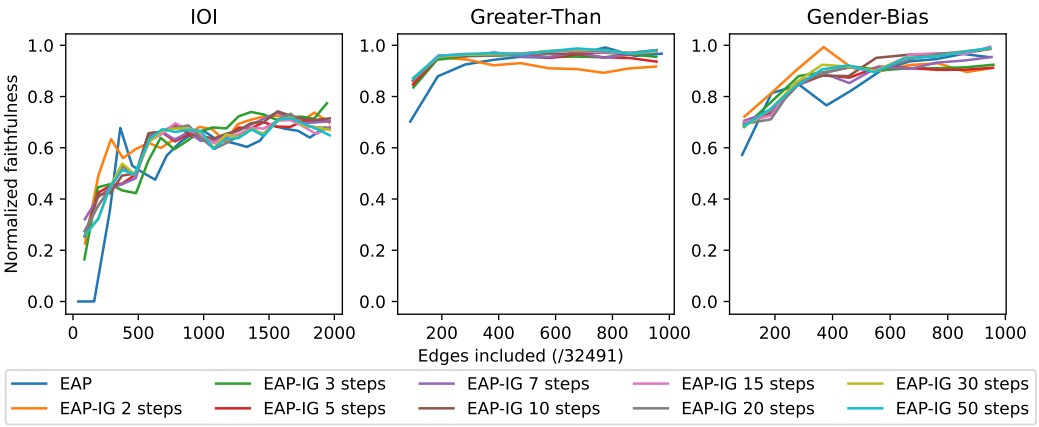

Figure 7: Faithfulness of circuits obtained using EAP-IG and varied step counts

## B  EAP-IG Circuit Overlap

In this section, we demonstrate that although our focus is on faithfulness, EAP-IG recovers the nodes and edge of manually found circuits as least as well as (and in fact better than) EAP does. We do so by taking scoring our model's edges using EAP and EAP-IG, and then, for $n = 1, \dots 200$, taking the top-$n$ edges as our circuit without pruning. We do this to replicate the methodology of Syed et al. (2023). We then compute the precision and recall of the EAP/EAP-IG circuits with respect to the manually found circuits. As the reference circuit for IOI excludes MLPs, so we also exclude this from our assessment.

Our results (Figure 6) indicate that EAP-IG does well at retrieving nodes and edges. On IOI, results for edge precision / recall are similar for EAP and EAP-IG. For node precision / recall, EAP-IG initially does better, though it slips slightly when recall grows high. Note that this assessment is likely somewhat flawed due to the ambiguous role of MLPs in the manually found IOI circuit. On Greater-Than, whose manual circuit also addresses MLPs, EAP-IG dominates in both the edge and node settings. It finds circuits with both better precision and better recall than EAP.

## C  EAP-IG Requires Few Steps

We test the number of steps required by EAP-IG on three tasks: IOI, Greater-Than, and Gender-Bias, since these tasks have distinct outcomes for EAP-IG. We do so by running EAP-IG for $m$ steps, for $m \in \{2, 3, 5, 7, 10, 15, 20, 30, 50\}$; note that at $m = 1$ step, EAP-IG is equivalent to EAP, which we plot as well. We then find circuits at various edge counts $n$.

Our results (Figure 7) indicate that few steps are needed to attain high faithfulness. At $m = 2$ steps EAP-IG produces an unfaithful circuits for Greater-Than, and mimics EAP's spike in faithfulness on the IOI task. All circuits found with $m > 2$ have similar, high, faithfulness across tasks. Although our results suggest even $m = 3$ steps would suffice, we use $m = 5$, to leave a margin for error. EAP-IG is thus 5 times slower than EAP; however, since this is only 5 forward and backward passes over the data, this is not a significant slowdown.

We note that integrated gradients typically requires a higher number of steps; Sundararajan et al. (2017) recommends between 20 and 300 to approximate the integral within 5%. We have two hypotheses that might explain this discrepancy. The first is that, for the purposes of EAP-IG, an accurate approximation is not very important; indeed, we see that EAP-IG does not accurately approximate activation scores despite its high faithfulness. But second, we note that the typical baseline used with integrated gradients is a zero baseline, e.g. a black image, in the case of computer vision tasks. It seems likely that activations of clean and corrupted inputs in our tasks are rather close together, while a regular image and black image are not. As a result, fewer steps may be needed.

## D   Comparing EAP-IG To Other Variants

Although we introduce one particular way of implementing EAP-IG in this paper, contemporary work has proposed many alternatives. In this section, we discuss a number of variants on EAP-IG, and then compare their performance on the tasks studied throughout this paper. We find that our version of EAP-IG outperforms other EAP-IG variants, though one baseline in particular is quite strong.

### D.1   EAP-IG Variants and Baselines

**EAP-IG in activation space (Marks et al., 2024)**   Marks et al. search for circuits of features, rather than nodes and edges. As a result, their conception of the transformer's graph is somewhat different from ours; they break the input embeddings, each block of attention heads, each MLP, and each point in the residual stream after an MLP, into a thousands of sparse features, each of which constitutes a node. Unlike our work, feature nodes are not connected to all downstream features nodes; instead, they are connected only to nodes in the next layer.

Marks et al. propose an EAP-IG method to score the edges of the thousands of feature nodes efficiently. However, to find the score of an edge $(u, v)$, they do not average the gradient when interpolating between different inputs with embeddings $z$ and $z'$, Rather, they take the gradient of $v$ when setting the *activations* of $u$ to a combination of its clean $u$ and corrupted outputs. Formally, they compute the score of an edge as:

$$(z'_u - z_u)\frac{1}{m}\sum_{k=1}^{m}\frac{\partial L(s|\text{do}(z_{u,out} = z_{u,corrupted} + \frac{k}{m}(z_{u,clean} - z_{u,corrupted})))}{\partial z_v}. \tag{4}$$

This quantity cannot be calculated with just $m$ forward passes; performing the *do* operation on multiple nodes that are not calculated in parallel will yield incorrect scores. As a result, Marks et al. iterate over sublayers that are calculated in parallel (the aforementioned input embeddings, attention head blocks, MLPs, and residual stream points), and perform $m$ forward passes for each. In the context of feature circuits, this is a significant improvement over ablating features individually; however, in the context of component / edge circuits, this scales worse than EAP-IG in input space.

**Partial EAP-IG in activation space (Miller et al., 2024)**   AutoCircuit, the circuit-finding framework introduced by Miller et al. (2024), introduces its own EAP-IG variant, called `mask_gradient_prune_scores`. Though Miller et al. do not discuss it directly, an analysis of their code suggests that this is indeed a distinct EAP-IG variant. They compute the score of an edge $(u, v)$ as roughly:

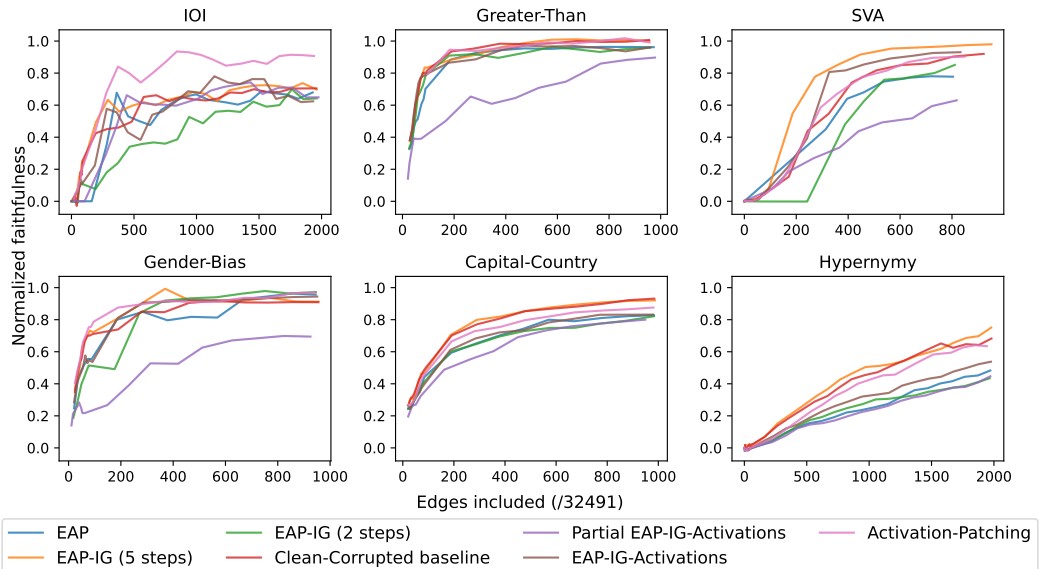

Figure 8: Faithfulness of circuits found using EAP-IG and variants.

$$(z'_u - z_u) \frac{1}{m} \sum_{k=1}^{m} \frac{\partial L(s|\text{do}(z_u = z'_u + \frac{k}{m}(z_u - z'_u)))}{\partial z_v}. \tag{5}$$

In contrast to the prior formulation, which set the output of each node to the interpolation between its clean and corrupted outputs, this formulation sets each node's output to an interpolation of its current and corrupted outputs. To calculate this quantity exactly, the *do* operation should only be performed in parallel on independent nodes, much like in the prior formulation; however, Miller et al. compute this in parallel for all nodes. Though the implementation we test is identical to theirs, our mathematical formulation of their procedure is thus imprecise. Letting $V$ be the set of all non-logit nodes in the model's computational graph, we could instead formulate it as follows:

$$(z'_u - z_u) \frac{1}{m} \sum_{k=1}^{m} \frac{\partial L(s|\text{do}(\forall n \in V : z_n = z'_n + \frac{k}{m}(z_n - z'_n)))}{\partial z_v}. \tag{6}$$

**Clean-Corrupted Baseline** In our definition of EAP-IG, we average gradients over an interpolation between clean and corrupted inputs. However, Appendix C suggests that the interpolated steps are not very important, as EAP-IG works with $m$ as low as 3. A reasonable baseline is thus EAP-IG, where the gradients used in scoring are just the average of the gradients on clean and corrupted inputs; this is distinct from EAP-IG when $m = 2$, as EAP-IG never calculates gradients using fully corrupted inputs.

$$(z'_u - z_u)^\top \left( \frac{1}{2} \nabla_v L(s) + \frac{1}{2} \nabla_v L(s') \right) \tag{7}$$

## D.2 Results

We run EAP, EAP-IG (with 2 and 5 steps), activation patching and all three methods mentioned above on all 6 tasks that we study, as in Section 4. Our results (Figure 8) show that our EAP-IG works roughly as well as Marks et al.'s slower EAP-IG variant across tasks. It also outperforms Miller et al.'s EAP-IG variant on all tasks but IOI. However, the Clean-Corrupted baseline is quite strong: EAP-IG outperforms it only on the SVA task. This

is not especially surprising, given how few steps EAP-IG needed to succeed, but EAP-IG with $m = 2$ steps does underperform EAP-IG with 5 steps and the Clean-Corrupted baseline. While our EAP-IG has a slight edge on SVA, the Clean-Corrupted baseline seems like a strong method in its own right, and certainly outperforms EAP.

# E   Circuit Search

After scoring edges using EAP or EAP-IG, we must select the set of edges that constitute our circuit. The naive, top-$n$ approach of Syed et al. (2023) is somewhat effective, but frequently returns circuits with parent- or childless nodes, which are then pruned. We can avoid this issue by using the following greedy algorithm:

**Find Circuit (Greedy)** (G, n)

$(V, E) \leftarrow G$
$C_V \leftarrow \{G.\text{logits}\}, C_E \leftarrow \{\}$
for $i = 1, \ldots, n$:
    $D \leftarrow \{e | e.\text{child} \in C_V, e \in E \setminus C_E\}$
    $e \leftarrow \arg\max_{e' \in D} |e'.\text{score}|$
    $C_E \leftarrow C_E \cup \{e\}$
    $C_V \leftarrow C_V \cup \{e.\text{parent}\}$
return $C_V, C_E$

This greedy algorithm is rather like a maximizing version of Dijkstra's algorithm. We initialize our circuit to include only the logits. Then, for each of $n$ steps, we collect every edge that is not in our circuit, but has a child node in our circuit. Of these, we select the edge with the highest absolute score. If its parent is not yet in the circuit, we add it too.

Because we only add nodes whose children are already in our circuit, we are guaranteed to never have childless nodes in our circuit. Running this same procedure in the reverse, starting from the input embeddings, would result in no parentless nodes. Finding the exact solution to this problem, i.e. finding the maximum-flow subgraph connecting our source node (the inputs) to the target node (the logits), is likely NP-hard.

Note that in both scenarios, we select edges based on the magnitude of their score, not its actual value. Selecting edges based on (positive) score would increase the circuit's performance, but the resulting circuit might omit components that are important in the way that they hurt model performance on the task. In the IOI task, for example, a number of heads play a significant role in reducing model performance on the task. Though those heads are negative, they still belong in our circuit, as they constitute a significant part of model behavior.

# F   Approximation Error and Kendall Correlations

How poorly do EAP and EAP-IG estimate the actual changes that occur upon corrupting a given edge? Syed et al. (2023) find that EAP estimates this quite poorly, and the same is true for EAP-IG. For each task, we compare the average difference between in score assigned by EAP/EAP-IG, and the actual score observed when performing activation patching. While the errors for EAP-IG are generally slightly smaller, the magnitude of these differences is much smaller than that of the errors themselves. Moreover, while the magnitude of the errors seems small, these errors add up over the computational graph's many edges.

How well do EAP's and EAP-IG's ranking of edges correlate with the ranking given by activation patching scores? Figure 9 indicate that EAP-IG indeed tends to have significantly worse correlations than EAP.

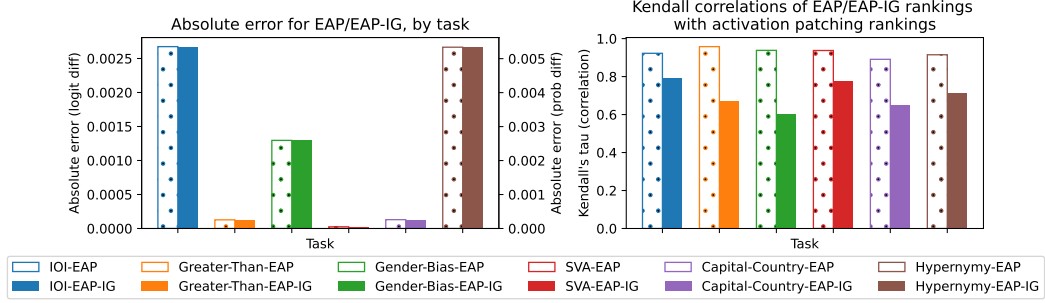

Figure 9: **Left**: Approximation error of EAP and EAP-IG, compared to the activation patching ground truth. **Right**: Kendall correlation between EAP/EAP-IG's edge rankings and the edge rankings from activation patching.

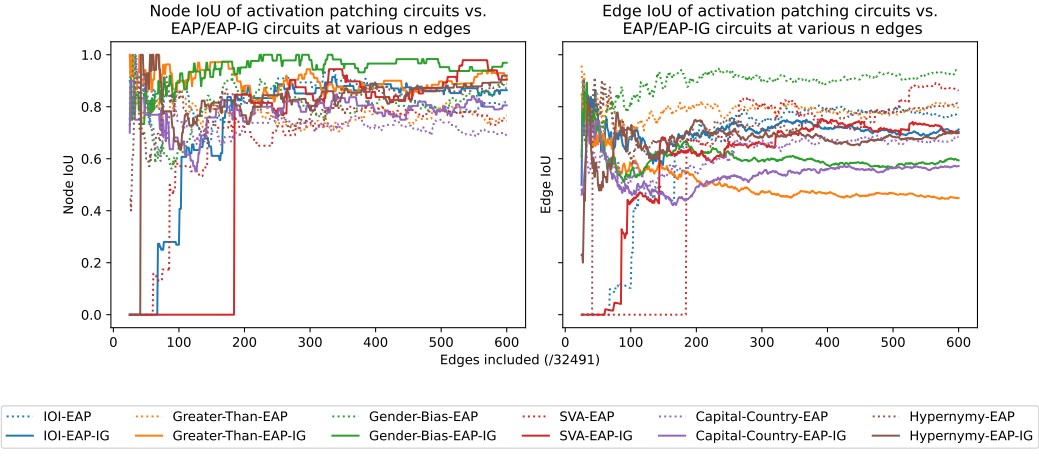

Figure 10: Node (left) and edge (right) intersection over union (IoU; Jaccard similarity)

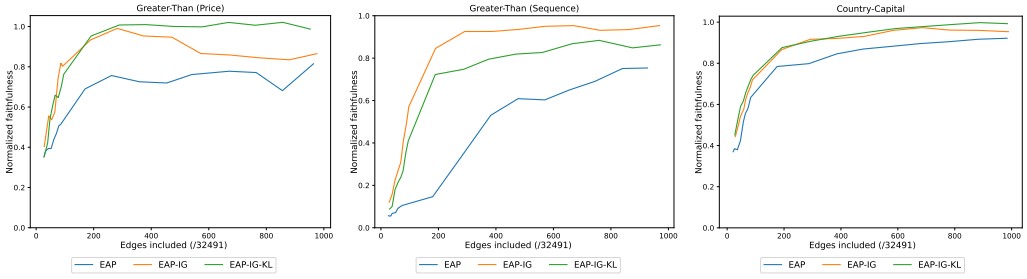

Figure 11: Faithfulness plots for Greater-Than (Price) Greater-Than (Sequence), and Country-Capital.

## G   Within-Task Overlap and Faithfulness

We have abundant evidence for the inability of overlap to predict faithfulness within (rather than across) tasks. We can show this precisely in the case of overlap with manually found circuits—the smallest EAP circuit for IOI that contains at least 90% of the manually-found IOI circuit's nodes achieves 0% faithfulness. The smallest EAP circuit for Greater-Than fulfilling that criterion does better, but is still only 51% faithful.

Our own experiments also provide evidence for overlap's unreliability: on SVA, the EAP circuit overlaps heavily with the activation patching circuit, but performs much worse, as shown in Figure 4 right. However, these experiments only measured the average overlap in edge rankings, not in circuit nodes / edges! It may well be that the edges being chosen for our circuits are not necessarily being chosen in the order of the ranking, due to our greedy algorithm and the fact that circuits are pruned prior to faithfulness evaluation.

In Figure 10, we compute node and edge overlaps at various edge counts $n$. Our previous statement does still hold: the node and edge overlap of the EAP circuit for SVA with the activation patching circuit for SVA, surpasses that of the EAP-IG circuit around $n = 250$. However, at $n = 250$, the EAP circuit's faithfulness is quite poor, while those of both EAP-IG and activation patching are quite good. This is likely because EAP is missing a particularly crucial edge, from the inputs to MLP 0.

## H   Circuit Overlap Significance Testing

We can compute the significance of the overlap of two circuits as follows. Imagine we have two circuits containing nodes $V_1, V_2$, and there are $N$ total nodes in the model's graph. Then we can use the hypergeometric distribution to model the probability of an overlap of at least size $k = |V_1 \cap V_2|$, given $n = |V_1|$ draws without replacement from a population of size $N$, of which $K = |V_2|$ contribute to overlap. The probability of an overlap of exactly size $k$ under a hypergeometric distribution is

$$\frac{\binom{K}{k}\binom{N-K}{n-k}}{\binom{N}{n}}. \tag{8}$$

The cumulative density function can similarly be used to be used to compute the desired quantity: the probability of an overlap of at least size $k$. We do this using SciPy (Virtanen et al., 2020).

## I   Greater-Than Variants

In this section, we discuss the two Greater-Than variants, Greater-Than (Price) and Greater-Than (Sequence), and the Country-Capital task. Before running our overlap and cross-task faithfulness experiments on them, we confirm that faithful circuits for them do indeed

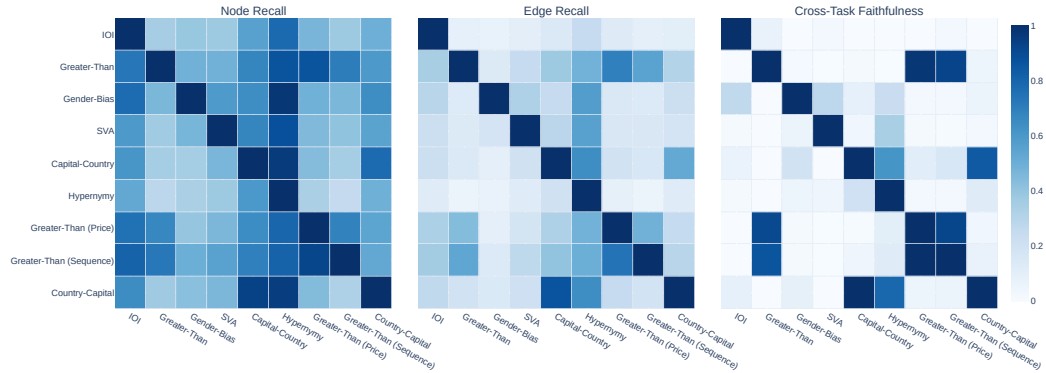

Figure 12: Node (left) and edge (center) recall, i.e. the fraction of nodes / edges from the y-axis task circuit that are also in the x-axis circuit. **Right**: Cross-task faithfulness. Each square is the faithfulness of the circuit from the y-axis task, tested on the x-axis task. Faithfulness is normalized, per test task, according to the faithfulness of the circuit for that task.

exist; in Figure 11, we see that this is the case. For the overlap and cross-task faithfulness experiments, we take the 300, 200, and 300 edge EAP-IG-KL circuits for Greater-Than (Price) and Greater-Than (Sequence), and Country-Capital respectively.

## J Asymmetric Overlap Measures

We noted previously that IoU cannot possibly predict faithfulness well, because faithfulness is a notably asymmetric relation. But could it be the case that asymmetric overlap measures might capture this quality that IoU missed? To test this, we take each pair $C_1, C_2$ of circuits, and measure the recall of $C_1$ on $C_2$ and vice-versa. Let $V_1$ and $V_2$ be the nodes of $C_1$ and $C_2$; then the node recall of $C_1$ on $C_2$ is $\frac{|V_1 \cap V_2|}{|V_2|}$. The definition for edge recall is analogous. We might hypothesize that if $C_1$'s recall on $C_2$ is high, its faithfulness on $C_2$'s task will be high as well.

The results of this experiment indicate that this is only sometimes the case. We can see that both node recall (Figure 12, left) and edge recall (Figure 12, center) are able to replicate some key trends of our faithfulness experiment (Figure 12, right, repeated from Figure 5 right). Besides the inter-Greater-Than similarity, they also capture the fact that Hypernymy covers many other circuits due to its large size—a fact reflected in its faithfulness. However, they introduce many new errors too. In particular, both measures seem to overestimate the similarity between pairs of tasks, such as Greater-Than (and its variants) and Hypernymy, among others. So, the recall metrics do not entirely resolve the issue of overlap vs. faithfulness. Although this does not preclude the existence of another asymmetric overlap metric that predicts faithfulness better, this indicates that at least the most obvious one does not work for this purpose.

## K Replications in Larger Models

We replicate our results in larger models, in order to show that EAP-IG is indeed still useful in such contexts, and that our findings do still hold. As before, we find circuits with EAP, EAP-IG, and EAP-IG-KL, and test their faithfulness. We omit activation patching circuits due to computational constraints. Similarly, we only test on GPT-2 XL (1.5B parameters; Radford et al., 2019) and Pythia-2.8B (Biderman et al., 2023), another decoder-only pre-trained language model with 2.8B parameters; due to the same constraints, we also evaluate faithfulness on only 100 examples for each task. Notably, the reason for these computational constraints is not EAP or EAP-IG; the slowest step in this process is the computation of

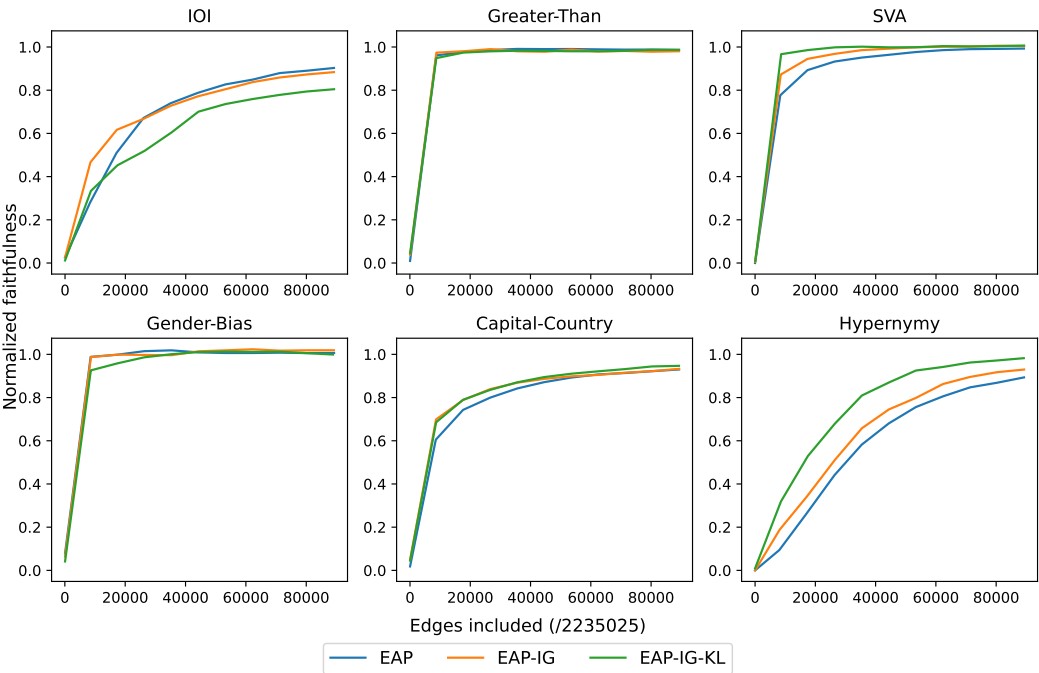

Figure 13: Faithfulness of circuits found via scores from EAP, EAP-IG, and activation patching for GPT-2 XL; values near 1.0 are better. EAP-IG circuits' faithfulness equals or surpasses EAP circuits'.

faithfulness, whose cost grows with the number of nodes in the circuit. Developing more efficient algorithms for computing faithfulness would be a useful direction for future work.

Our results on both GPT-2 XL (Figure 13) and Pythia-2.8B (Figure 14) confirm our earlier findings. In both models, when there are discernable differences between the faithfulness of circuits found, EAP-IG performs better than EAP. As before, the hypernymy circuit is especially hard to find. In larger models, the total failure case of EAP-IG, as in SVA in GPT-2 small, seem to have disappeared, although we do not test faithfulness at very low edge counts as in our main experiments.

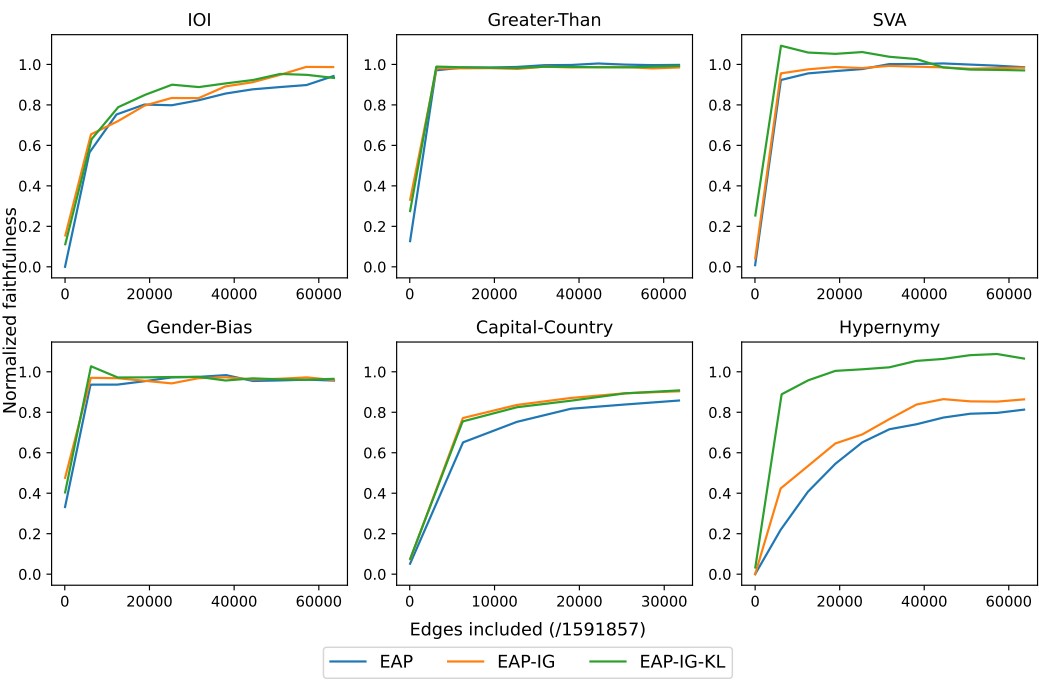

Figure 14: Faithfulness of circuits found via scores from EAP, EAP-IG, and activation patching for Pythia-2.8B; values near 1.0 are better. EAP-IG circuits' faithfulness equals or surpasses EAP circuits'.

