# OpenReview forum: "Have Faith in Faithfulness: Going Beyond Circuit Overlap When Finding Model Mechanisms"
_colmweb.org/COLM/2024/Conference — COLM_

### Official Review · Reviewer_vH5S · 2024-05-08

**Rating:** 8
**Confidence:** 5
**Ethics Flag:** 1

**Summary:**

Edge attribution patching (EAP) is a circuit discovery technique approximating the impact of causal interventions on model edges via gradient attribution. Its effectiveness is commonly validated by comparing the overlap of its resulting circuits with those found by activation patching methods with causal guarantees, but rarely in terms of faithfulness, i.e. whether behavior of the model remains consistent after all non-circuit edges are ablated.

This work proposes an extension to edge attribution patching, EAP-IG, using integrated gradients to improve the faithfulness of the resulting circuits. The faithfulness EAP, EAP-IG and activation patching is evaluated, showing that no-overlap and full-overlap between EAP-like methods and activation patching are generally good indicators of unfaithful and faithful (respectively) circuit identification. On the contrary, circuits with moderate overlap cannot generally assumed to be faithful to model behavior, pointing to the importance of faithfulness evaluation in this setting.

**Reasons To Accept:**

- The proposed method and related analysis in the paper are explained thoroughly and suggest a promising way forward in the evaluation of circuit discovery approaches.
- The analysis of disagreement between overlap and faithfulness metrics of circuit quality provides a valuable perspective, identifying existing limitations in current evaluation practices for attribution patching approximations.
- The proposed EAP-IG is a refreshing application of an established feature attribution method for model component attribution. Notably, it allows for different gradient propagation targets and is convincingly shown to be more faithful than EAP.
- An evaluation is conducted to ensure the findings generalize across several tasks and model sizes.

**Reasons To Reject:**

- Minor: the name of the model evaluated in the main body of the paper is never mentioned explicitly, although I assume it is GPT-2 small based on the task section.
- While results discussed in Appendix C suggest that a low number of interpolation steps might be sufficient to obtain good faithfulness performances, this estimate is based on a set of toy tasks. There is no guarantee that such property would hold in case of more complex tasks.
- More generally, as for other circuits discovery approaching requiring corrupted examples, EAP-IG is limited by the need of such pre-defined pairs to function. Provided that several tasks among those used here admit several possible corrupted variants, ideally a broader set of baselines should be evaluated for every example, and confidence intervals should be reported to confirm the robustness of the proposed method.

---

> ### Author Rebuttal · Authors · 2024-05-30
>
> Thank you for your insightful review! We’re happy to hear that you found our paper thorough and our perspective valuable. Here are our responses to your comments:
> - Thanks for pointing out that we don’t explicitly mention the model we use in the paper; we will update it to more directly state that we run our experiments on GPT-2 small.
> - True—the number of steps needed for a good approximation depends on the loss landscape of your task. While 5 is sufficient for the tasks we study, different tasks could have different landscapes; we’ll add a note to this effect.
> - This is an interesting idea! For some tasks (e.g. hypernymy or fact-retrieval the corruption is randomly sampled from multiple candidates: “Paris, the capital of [France]” could just as easily be corrupted with (Rome, Italy) as with (Brussels, Belgium). Using different corrupted variants and computing confidence intervals seems like a good idea; we’ll try to add this sort of analysis.

---

> > ### Comment · Reviewer_vH5S · 2024-06-05
> >
> > Thank you for your response; I am glad you found my suggestions helpful and that you will add details/notes about the first two points and possibly some extra analysis for the third point. I do not have further questions, and I find my initial rating still applies in light of these changes.

---

### Official Review · Reviewer_nt6V · 2024-05-10

**Rating:** 6
**Confidence:** 4
**Ethics Flag:** 1

**Summary:**

The paper introduces a new method called Edge Attribution Patching with Integrated Gradients (EAP-IG), which aims to efficiently identify circuits in language models while ensuring these circuits are faithful. EAP-IG, which combines EAP with IG to better assess edge importance in circuits, especially addressing the issue of zero gradients which can misrepresent an edge's importance by extrapolating effects between two points using a linear expansion. The main results show that circuits found using EAP-IG yield with high faithfulness as defined in the paper. Additionally, the authors also explore the circuit overlap between tasks.

**Reasons To Accept:**

- The research topic, to faithfully scale circuit discovery method, is very timely. The paper extends previous circuits discovery methods EAP by taking inspirations from IG. The proposed method EAP-IG better assesses edge importance in circuits, especially addressing the issue of zero gradients which can misrepresent an edge's importance by extrapolating effects between two points using a linear expansion.
- The paper is easy to follow, and study the topic both theoretically and also empirically. The authors also show circuits similarity between tasks, which is quite novel. The paper found that similar tasks often come with more overlapping circuit components.

**Reasons To Reject:**

- The definition of faithfulness could be more precisely grounded. The faithfulness described in the paper refers specifically to whether the identified circuit comprises a complete set of components that control model behaviors, **conditioned on** zero-ablate or mean-ablate interventions. However, this circuit might not be the definitive one. Furthermore, iiuc, there is a potential "hydra-effect," where multiple circuits exist, and one circuit may only become apparent due to interventions.

- Section 4.4 could be clearer. I am not an expert on this, but it feels somewhat counterintuitive to see that vanilla activation patching could outperform gradient-based methods in terms of improving faithfulness—that is, the completeness of the identified circuits with respect to task performance. The analyses of overlap between different patching methods (gradient-based or intervention-based) might also suggest that numerous types of circuits exist, which makes the claim that any one of them is more faithful seem unfounded. Could the authors clarify this?

---

> ### Author Rebuttal · Authors · 2024-05-30
>
> Thanks for your thorough review! We’re glad you found the paper easy to follow and timely. The weaknesses you mention touch on developing areas of circuit analysis; we respond here:
>
> **Faithfulness**: Our definition of circuit faithfulness is well-grounded in prior literature: Wang et al. (2023) define it as whether “the circuit can perform the task as well as the whole model”; Hanna et al.’s (2023) usage is similar. However, we agree that there are many ways to operationalize faithfulness. We follow Wang et al., Hanna et al. and Conmy et al. (2023) in using activation patching (AP), but other work uses mean ablations, gaussian noising, or even zero ablations.
>
> Thus, we agree that there may be variance in the circuits found by each method. However, AP is a standard method employed by a great deal of prior work. Moreover, evidence suggests that AP finds circuits similar to mean ablations; Wang et al. repeated their experiments with both techniques and found the same IOI circuit. While AP may not yield the same results as e.g. noising or zero ablations, not all methods are equally valid: [Zhang and Nanda (2024)](https://openreview.net/forum?id=Hf17y6u9BC) argue against noising, and many others against zeroing.
>
> **Completeness**: Your question about “whether the identified circuit comprises a complete set of components” corresponds to what Wang et al. call completeness. Tests of completeness are scarce; Wang et al. test it with randomized patching, but this is costly. Fortunately, while overlap with a reference is a poor metric for faithfulness, it is a reasonable measure of completeness, and EAP-IG circuits capture the published IOI and Greater-Than circuits better than EAP (App. B, Fig. 6). Note that the contrast between faithfulness and overlap/completeness is a major contribution of this work.
>
> **Section 4.4**: To clarify, gradient-based methods like EAP/-IG approximate the ground-truth edge importance scores given by AP, which are then used to find a circuit. It is thus expected that AP outperforms them.
>
> We’re not sure that numerous types of circuits exist, but the existence of multiple reasonable circuits wouldn’t undermine our claims re: faithfulness. We directly measure faithfulness, and EAP-IG circuits are more faithful than EAP. It’d be noteworthy if our circuits were less complete, but App. B/Fig. 6 suggest this is not the case.
>
> We’re out of space now, but are happy to explain more! If accepted, we’ll add more discussion of these three points.

---

### Official Review · Reviewer_ekio · 2024-05-10

**Rating:** 7
**Confidence:** 3
**Ethics Flag:** 1

**Summary:**

This paper presents a simple adaptation to edge attribution patching (EAP) for scoring edges of circuits in an LLM by borrowing the notion of integrated gradients by Sundararajan et al. 2017; doing so allows for the extraction of more faithful circuits when compared to EAP performing on a par or even better that the more expensive non-approximate method of activation patching circuits. The authors also do a study on the correlation between overlap of circuits (nodes, edges) and the faithfulness which doesn't necessarily lead to a causal relationship. The evaluation is done using GPT-2 small (although the appendix has a couple more experiments using GPT-2 XXL and Pythia-3.8B) on 6 corruption tasks.

**Reasons To Accept:**

* The contribution of this paper is not so much on the novelty of the proposed algorithm but on the simplicity, elegance and effectiveness of it. Just to be clear, I am not suggesting that the produced algorithm isn't novel (on the contrary!) but that the outcome essentially stems from an incremental observation of previous works. Having said that I believe that since there is a substantial improvement wrt faithfulness it's an addition to the mechanistic interpretability community that will be received positively.

* I believe that the more significant contribution stems from the extensive analysis and experiments conducted to verify that EAP-IG works and why. The overlap vs faithfulness findings were also interesting and I appreciated the pragmatic and somewhat cautious view that overlap isn't always a good predictor of faithfulness across tasks.

* For someone who is not an expert on mechanistic interpretability, Section 2 is a very nicely written introduction to the field; much appreciated!

**Reasons To Reject:**

* I found the extra Greater-Than tasks that were constructed to verify the existence of robust circuits that appear across tasks to be somewhat obvious and derived from the simplest of all tasks in my opinion. Personally, I found the relationship between hypernymy and the country-capital tasks a lot more interesting (and fascinating!), so I wonder why the authors didn't pursue constructing similar derivative tasks with sub-categories that have expert-defined similarities, e.g., countries that are nearby, hypernyms that are farther/closer to each other according to Wordnet etc.

---

> ### Author Rebuttal · Authors · 2024-05-30
>
> Thank you for your positive review! We appreciate your enthusiasm for our paper, and are particularly happy that you found Section 2 to be a good introduction to circuits work—making this work accessible to people in and outside the mechanistic interpretability community was a priority for us.
>
> About the use of Greater-Than variants: we want to clarify that these tasks were studied previously in [Hanna et al. (2023; Appendix H)](https://arxiv.org/abs/2305.00586), and were already known to have high cross-task overlap and faithfulness. We studied them for this very reason: we had discovered instances of moderate-high cross-task overlap leading to low faithfulness (hypernymy & country-capital) and wanted to also provide examples of moderate-high overlap leading to high faithfulness (Greater-Than variants).
>
> We appreciate the suggestion to focus more on hypernymy and fact-retrieval—we found them interesting as well, and actually did study fact-retrieval in greater detail! We took the original fact-retrieval task (“Paris, the capital of [France]”) and constructed a task that queried the reverse relation (“France, whose capital, [Paris]”). We found that the circuits for the fact-retrieval task and its reverse have high overlap and high cross-task faithfulness with one another! This is a little surprising in light of findings like the [reversal curse](https://arxiv.org/abs/2309.12288), which suggests that models’ knowledge relations are not always invertible, at least in in-context learning scenarios. We didn’t think to include these results in our submission, but we’d be happy to update the paper to do so if accepted.

---

> > ### Comment · Reviewer_ekio · 2024-06-05
> >
> > Thanks a lot for your response and the short discussion on the fact-retrieval task and the surprising results. I believe it could make a nice addition to the paper's results given the nature of the task being different from the rest.

---

### Decision · Program_Chairs · 2024-07-10

**Decision:**

Accept

**Comment:**

This paper introduces a variant of the edge attribution patching (EAP) method for circuit discovery that uses integrated gradients (EAP-IG). The authors show that this adaptation finds a more faithful circuit compared to EAP. Faithfulness is defined in this paper with respect to the baseline circuit found by causal activation patching. They also study the relationship between neuron/edge overlap and behavioral similarities across circuits. The study primarily focused on GPT-2-small with a few language-like and arithmetic tasks but also included a preliminary analysis with a larger GPT-2 variant. An additional task of bi-directional factual associations was briefly mentioned in the discussion period and was received positively.

The paper is well-written and of high quality; the topic of scalable circuit discovery is timely. The approach is simple yet effective, which reviewers found “elegant” and “a refreshing application of an established feature attribution method.” Reviewers also found the experiments to be thorough and the work to be interesting from both theoretical and experimental perspectives. There were concerns about how faithfulness was quantified, which authors attempted to address in the discussion, providing a good summary of different variants of ablations used as proxies for faithfulness.